# Composite Interventions on Outcomes of Severely and Critically Ill Patients with COVID-19 in Shanghai, China

**DOI:** 10.3390/microorganisms11071859

**Published:** 2023-07-23

**Authors:** Jiasheng Shao, Rong Fan, Chengnan Guo, Xuyuan Huang, Runsheng Guo, Fengdi Zhang, Jianrong Hu, Gang Huang, Liou Cao

**Affiliations:** 1Department of Immunology and Rheumatology, Jiading District Central Hospital Affiliated Shanghai University of Medicine & Health Sciences, Shanghai 201899, China; shaojiasheng@jdhospital.com; 2Tulane National Primate Research Center, Tulane University School of Medicine, 18703 Three Rivers Road, Covington, LA 70433, USA; rfan1@tulane.edu; 3Genomics, Biotechnology Center, Center for Molecular and Cellular Bioengineering, Technische Universität, 01307 Dresden, Germany; 4Shanghai Institute of Infectious Disease and Biosecurity, School of Public Health, Fudan University, Shanghai 200032, China; cnguo22@m.fudan.edu.cn; 5Department of Urology, Renji Hospital, Shanghai JiaoTong University, Shanghai 200127, China; huangxuyuan@jdhospital.com; 6Department of General Surgery, Jiading District Central Hospital Affiliated Shanghai University of Medicine & Health Sciences, Shanghai 201899, China; guorunsheng@jdhospital.com; 7Department of Infectious Disease, Shanghai East Hospital, Tongji University School of Medicine, Shanghai 200120, China; zhangfengdi@126.com; 8Department of Respiratory Medicine, Jiading District Central Hospital Affiliated Shanghai University of Medicine & Health Sciences, Shanghai 201899, China; hujianrong@jdhospital.com; 9Shanghai Key Laboratory of Molecular Imaging, Shanghai University of Medicine and Health Sciences, Shanghai 201318, China; 10Department of Nephrology, Molecular Cell Lab for Kidney Disease, Shanghai Peritoneal Dialysis Research Center, Ren Ji Hospital, Uremia Diagnosis and Treatment Center, Shanghai Jiao Tong University School of Medicine, Shanghai 200127, China

**Keywords:** COVID-19, Azvudine, Paxlovid, interleukin-6 receptor antagonist, baricitinib, α-thymosin, intravenous immunoglobulin

## Abstract

**Background**: The sixty-day effects of initial composite interventions for the treatment of severely and critically ill patients with COVID-19 are not fully assessed. **Methods**: Using a Bayesian piecewise exponential model, we analyzed the 60-day mortality, health-related quality of life (HRQoL), and disability in 1082 severely and critically ill patients with COVID-19 between 8 December 2022 and 9 February 2023 in Shanghai, China. The final 60-day follow-up was completed on 10 April 2023. **Results**: Among 1082 patients (mean age, 78.0 years, 421 [38.9%] women), 139 patients (12.9%) died within 60 days. Azvudine had a 99.8% probability of improving 2-month survival (adjusted HR, 0.44 [95% credible interval, 0.24–0.79]), and Paxlovid had a 91.9% probability of improving 2-month survival (adjusted HR, 0.71 [95% credible interval, 0.44–1.14]) compared with the control. IL-6 receptor antagonist, baricitinib and a-thymosin each had a high probability of benefit (99.5%, 99.4%, and 97.5%, respectively) compared to their controls, while the probability of trail-defined statistical futility (HR > 0.83) was high for therapeutic anticoagulation (99.8%; HR, 1.64 [95% CrI, 1.06–2.50]) and glucocorticoid (91.4%; HR, 1.20 [95% CrI, 0.71–2.16]). Paxlovid, Azvudine, and therapeutic anticoagulation showed a significant reduction in disability (*p* < 0.05) **Conclusions**: Among severely and critically ill patients with COVID-19 who received 1 or more therapeutic interventions, treatment with Azvudine had a high probability of improved 60-day mortality compared with the control, indicating its potential in a resource-limited scenario. Treatment with an IL-6 receptor antagonist, baricitinib, and a-thymosin also had high probabilities of benefit in improving 2-month survival, among which a-thymosin could improve HRQoL. Treatment with Paxlovid, Azvudine, and therapeutic anticoagulation could significantly reduce disability at day 60.

## 1. Introduction

At the end of 2022, China was inclined to change course and adopt a “living with COVID-19” strategy [1]. Since then, COVID-19 infections have spread rapidly in major cities in China, including Shanghai, where the predominant SARS-CoV-2 variant, Omicron BF.7, has put great pressure on healthcare facilities. A model foresees that China’s outbreak will reach a first peak on 13 January 2023, with 3.7 million new cases per day, and COVID-19-related deaths are expected to peak 10 days later at around 25,000 per day [2].

To date, the majority of studies have mainly focused on the clinical characteristics and shorter-term progression of the viral infection in mild to moderate cases [3,4]. Huang et al. have described the one- and two-year evolution of health outcomes in COVID-19 survivors, regardless of initial disease severity [5,6]. However, few studies with large sample sizes have specifically reported the clinical outcomes of severely and critically ill survivors of COVID-19. In severe and critical cases, cytokine storm is believed to be one of the major reasons for acute respiratory distress syndrome (ARDS) and multiple-organ failure, which means comprehensive interventions should be administered in the early stage of the infection to reduce mortality [7]. Some cohort research indicated critically ill patients who received antiviral agents, immune modulators, immunoglobulin, anticoagulation, antiplatelet, and corticosteroid therapy had broad variations in the clinical outcome [8,9,10,11,12,13]. For instance, antiviral agents such as nirmatrelvir-ritonavir (Paxlovid) and Molnupiravir are currently available for the treatment of COVID-19 infection. These medications have been shown to reduce the risk of mortality in the post-acute phase in hospitalized patients [14].

However, limited data is available regarding the initial comprehensive interventions translating into clinical effects on survival, disability, and health-related quality of life (HRQoL), particularly for severely and critically ill patients with COVID-19 [5]. The aim of the study is to evaluate the effectiveness of these composite treatments on 60-day outcomes, including mortality, disability, and health-related quality of life (HRQoL), for patients who receive one or more of these treatments.

## 2. Methods

### 2.1. Study Designs and Participants

During the COVID-19 pandemic, we conducted a retrospective, single-centered study involving 1082 severely and critically ill patients with COVID-19, which were confirmed by reverse transcriptase polymerase chain reaction (RT-PCR) or COVID-19 antigen testing between 8 December 2022 and 9 February 2023 in Shanghai, China. The severe and critical illness of COVID-19 infection was defined by Guidelines on the Diagnosis and Treatment of COVID-19 (10th Trial Edition) [15]. The final 60-day follow-up was completed on 10 April 2023.

The study was approved by the Ethics Committee of Jiading District Central Hospital Affiliated Shanghai University of Medicine & Health Sciences and performed (Approval code: 2023K15). Patients or their lawful caretakers provided written informed consent. The demographic profile (age, sex, pre-existing disorders) for enrolled participants, clinical course (diagnosis date, admission date, symptoms, treatment recipes, disease severity), and vaccination dose information were extracted from medical records.

We divided all patients who received one or more interventions into six therapeutic domains: antivirals, immune modulators, intravenous immunoglobulin, antiplatelet, anticoagulation, and glucocorticoids, which were based on a previous study and our national treatment guideline [9,15]. In this retrospective study, each domain represents an investigation of specific treatment effectiveness. For example, patients who met the inclusion criteria of the antiviral domain were included in either the nirmatrelvir-ritonavir (Paxlovid) group, the Azvudine (a nucleoside analog) group, or the control group, whereas patients who met the exclusion criteria were not assigned to the antiviral domain. Briefly, patients in the antiviral domain received Paxlovid, Azvudine, or no antiviral medicines; patients in the immune modulation domain received tocilizumab (an IL-6 receptor antagonist), baricitinib (a Janus kinase-JAK inhibitor), α-thymosin (a non-specific T cell activator), or no immune modulator; patients in the intravenous immunoglobulin domain receive immunoglobulin for 3–5 days (given if clinical deterioration occurs); patients in the antiplatelet domain received aspirin, a P2Y12 inhibitor (clopidogrel or ticagrelor), or no antiplatelet therapy; patients in the anticoagulation domain received thrombo-prophylactic or therapeutic-dose anticoagulation with low molecular weight heparin (LMWH) or in accordance with usual administration; patients in the glucocorticoid domain received a (7–10)-day course of intravenous dexamethasone or methylprednisolone, or no glucocorticoids [15].

The medications used in each domain were as follows: Paxlovid (nirmatrelvir 300 mg and ritonavir 100 mg twice per day for 5 consecutive days), Azvudine (5 mg once per day for up to 14 days), tocilizumab (8 mg/kg per day for 2 doses), baricitinib (4 mg per day for up to 14 day), α-thymosin (1.6 mg once per day for at least 7 consecutive days), intravenous immunoglobulin (20 g per day for up to 5 days), antiplatelet (aspirin 100 mg once daily, clopidogrel 75 mg once daily, or ticagrelor 60 mg twice daily), anticoagulation (thrombo-prophylactic LMWH 4000 IU per day or therapeutic dose 100 IU/kg twice per day), and fixed-dose glucocorticoid (dexamethasone, 5 mg per day or methylprednisolone 40 mg per day) [12,15,16]. The flowchart of our cohort study is shown in Figure 1. The baseline characteristics of patients in one or more domains enrolled are presented in Table 1. Domain-specific inclusion and exclusion criteria for patients within each domain are shown in Table 2.

### 2.2. Outcome Measures

The main outcome of our study was to assess all-cause mortality within 60 days. The secondary outcome included HRQoL at 60 days measured using the 5-level EuroQol-5 Dimension (EQ-5D-5L) utility score and visual analog scale (VAS) score, and disability level at 60 days measured using the 36-item World Health Organization Disability Assessment Schedule (WHODAS) 2.0 [17]. Data was collected by face-to-face or telephone interview with the participants, their relatives, or a health care professional in our hospital.

The EQ-5D-5L is a preference-based health-related quality of life (HRQoL) instrument comprised of five dimensions: mobility, self-care, usual activities, pain/discomfort, and anxiety/depression. Respondents are asked to choose the most appropriate option from five alternatives (no, slight, moderate, severe, or extreme problems). Additionally, respondents were asked to indicate their present health state on a visual analogue scale (EQ VAS) ranging from the worst imaginable health state (“0”) to the best imaginable health state (“100”). EQ-5D-5L utility scores were calculated where a valid response (0 to 4) was available for each of the 6 EQ-5D-5L domains. Scores were calculated using the crosswalk link function and the individual responses to the EQ-5D-5L descriptive system, using the China time trade off (TTO) value set, with values between −0.391 and 1.0 [18].

The 36-item WHODAS 2.0 covers six domains of functioning with scores ranging from 0 (no difficulty) to 4 (extreme difficulty) and a total score ranging from 0 to 144, with higher scores representing greater disability. The total score is divided by 144 and multiplied by 100 to convert it to a percentage of maximum disability. WHODAS percentage scores were used to determine mutually exclusive disability categories: (1) no disability (0–4.5%); (2) mild disability (4.5–24.5%); (3) moderate disability (24.5–49.5%); (4) severe disability (49.5–95.5%); and (5) complete disability (95.5–100%) [17].

### 2.3. Statistical Analysis

The primary analysis was performed using a Bayesian piecewise exponential model. The underlying hazard rate was piecewise constant for each 10-day period up to day 30 and the 30-day period from day 30 to day 60. The prior distribution for each hazard rate was a γ distribution with 1 day of exposure and a mean equal to the total exposure (in days) divided by the total number of events. The primary model estimated treatment effects (log hazard ratios [HRs]) for each intervention relative to control within each domain with standard normal priors. The primary model included variables for each domain with each domain treatment as a category (with control interventions from each domain set as the referent) and was adjusted for patient age (categorized into 4 groups), sex, smoking, disease severity, preexisting conditions, respiratory supports, and other treatments from other domains. The posterior distributions of the interventions’ HRs were summarized with medians, 95% credible intervals (CrIs), and the probability that an intervention was superior to the control for that domain (i.e., HR < 1.0). Harm was defined as the probability that the HR was greater than 1. Futility was defined as the probability that there was not more than a 20% relative improvement in outcome (HR > 0.83). A prespecified interaction was modeled between antiplatelet therapy (pooled P2Y12 inhibitor and aspirin group) in the antiplatelet domain and therapeutic-dose heparin in the anticoagulation domain. Statistical thresholds based on posterior probabilities for superiority and harm were used for the primary outcome to determine trial stopping rules but were not used to guide the interpretation of other findings; rather, effect sizes along with posterior probabilities are presented for all analyses.

Sixty-day mortality was analyzed with a Bayesian logistic regression model. The EQ-5D-5L utility score was analyzed with a 2-part/mixture model including 2 components: a continuous distribution of EQ-5D-5L utility scores for patients who survived to day 60 and a point mass of 0 for patients who died before day 60. The posterior distributions of the mean difference between treatment and control for EQ-5D-5L utility scores were summarized with medians, 95% CrIs, and the probability that an intervention was superior to the control for that domain (i.e., a mean difference less than 0). Treatment effects were estimated for all patients, along with estimates for survivors only. HRQoL assessments were conducted for patients who survived during the 60-day follow-up period. Given the relatively short duration of the study, HRQoL data were collected from all surviving patients, excluding those who were deceased. To account for the impact of the deceased population on the analysis, we employed a 2-part/mixture model. The EQ VAS score was reported using descriptive statistics only, and the WHODAS disability category was calculated with ordinal mixture mode [17].

## 3. Results

In the main outcome of our study, we found that 139 out of 1082 patients (12.9%) died within 60 days. Figure 2 and Figure 3 illustrate that Azvudine had a higher probability of benefit compared to their control group (99.8%), while Paxlovid had a probability of benefit of 91.9%. IL-6 receptor antagonist, baricitinib and a-thymosin each had a higher probability of benefit (99.5%, 99.4%, and 97.5%, respectively) compared to their control groups. On the other hand, the probability of benefit for intravenous immunoglobulin, therapeutic anticoagulation, antiplatelet, and glucocorticoid compared to their control groups was 52.5%, 1.5%, 90.3%, and 26.1%, respectively. However, the probability of trail-defined statistical futility was high for therapeutic anticoagulation (99.8%; HR, 1.64 [95% CrI, 1.06–2.50]), glucocorticoid (91.4%; HR, 1.20 [95% CrI, 0.71–2.16]). More detailed information regarding the parameters is provided in Figure 3.

In the secondary outcomes, we assessed the EQ-5D-5L utility score, EQ VAS, and WHODAS 2.0 score in 941 out of 943 survivors in follow-up (99.8%). In the overall observation, the median (IQR) EQ-5D-5L utility score in survivors was 0.64 (0.12–0.89) (*n* = 941), and the median (IQR) EQ VAS score was 70 (55–80) (*n* = 941). The mean EQ VAS scores in each domain are shown in Table 3. Of the 941 survivors, 516 (54.83%) had moderate, severe, or complete disabilities at day 60. Notably, among these interventions, Paxlovid, Azvudine, and therapeutic anticoagulation showed significant reductions in disability (*p* < 0.05). Detailed information on disability among survivors is available in Table 4 and Table 5.

The adjusted mean difference in the EQ-5D-5L utility score in the α-thymosin group was 0.08 (95% CrI, 0.01–0.16) units higher compared with the control, with a posterior probability of superiority of 97.7%; among all patients (survivors and non-survivors), the adjusted mean difference in the EQ-5D-5L utility score was also 0.08 (95% CrI, 0.00–0.16), with a probability of superiority of 97.6% (Figure 4). Based on the results, the mean EQ-5D-5L utility score in the Paxlovid group was lower than the control group, and the posterior probability of harm was 100.0% among survivors and all patients. Similarly, the mean EQ-5D-5L utility score in the Azvudine group was also lower compared to the control group, with a posterior probability of harm of 95.1% among survivors and 95.9% among all patients. The mean difference in EQ-5D-5L utility scores between each remaining intervention and their control group is presented in Figure 4.

## 4. Discussion

In real-world practice, clinical interventions are administered to severely and critically ill patients with the aim to increase long-term survival and improving HRQoL and functional status for survivors. However, most clinical studies in severely and critically ill patients have assessed shorter-term outcomes, which may not be patient-centered [19,20]. Longer-term trends after severe and critical illnesses such as hypoxemia, acute respiratory distress syndrome (ARDS), and septic shock are characterized by frequent re-admission, persistent impairments in HRQoL and functional status, an elevated risk of mortality, and worsening of pre-existing chronic disorders that may last for years following initial admission. Additionally, patients who initially survive may face late morbidity and mortality risks that perhaps outweigh the benefits of treatment. Therefore, the effectiveness of many regimens utilized in patients with severe and critical illnesses, including COVID-19, remains unclear.

Paxlovid is recommended for mild to moderate COVID-19 cases within 5 days of symptom onset [15]. Liu et al. confirmed that it can be administered safely in severe adult patients with SARS-CoV-2 infection but did not significantly reduce 28-day all-cause mortality or the duration of SARS-CoV-2 RNA clearance in these patients [21]. Azvudine, a nucleoside analog that inhibits human immunodeficiency virus (HIV)-1 RNA-dependent RNA polymerase (RdRp), has also shown effectiveness in treating patients with COVID-19 [22]. Our study indicated Azvudine had a 99.8% probability of improving survival over two months in COVID-19 patients with severe and critical illness when compared with the control. However, it did not improve HRQoL in survivors. Meanwhile, although Paxlovid showed a 91.9% superiority in terms of survival, the follow-up duration was not long enough to draw definitive conclusions.

Previous studies proved that IL-6 can activate the Janus kinase-signal transducer and activator of transcription (JAK-STAT) pathway, induce an inflammatory response, and possibly form a cytokine storm, which is an important factor for the development of ARDS and extrapulmonary organ damage, and that an IL-6 receptor antagonist can suppress the over-activation of the human immune system [7]. JAK inhibitors can reduce the inflammatory response induced by infection. A recent pilot study confirmed that treatment with baricitinib plus standard of care (including use of corticosteroids) in critically ill patients with COVID-19 who were receiving invasive mechanical ventilation (IMV) or extracorporeal membrane oxygenation (ECMO) can reduce all-cause mortality at 28 days and 60 days [16]. The two medicines are also recommended by our national guideline for the treatment of severely and critically ill patients with COVID-19 [15,23]. In this observation cohort, both the IL-6 receptor antagonist and baricitinib had a higher probability of improving survival over 2 months.

Lymphocytopenia is a strong indicator of disease severity and prognosis in COVID-19 patients [24]. Liu et al. reported that treatment with α-thymosin can significantly reduce mortality in severe COVID-19 patients with severe lymphocytopenia by increasing T cell numbers and reversing T cell exhaustion [25]. However, the therapeutic potential of α-thymosin remains controversial, with some studies reporting paradoxical results. In our investigation, α-thymosin, as a non-specific T cell activator, demonstrated a higher probability of improving both survival and HRQoL over 2 months. The observed discrepancies between studies could be attributed to differences in disease severity among patients [26].

Conversely, other treatment domains, including intravenous immunoglobulin and therapeutic anticoagulation, were found to be ineffective in improving patients’ 60-day mortality rates. It has been demonstrated that intravenous immunoglobulin has multifunctional immunomodulatory properties such as inhibiting the activation of the complement and proliferation of T helper 17 cells, neutralizing auto-antibodies, impairing the antigen presenting capabilities of dendritic cells, and expanding regulatory T cell populations. Therefore, it could be a good therapeutic strategy for hospitalized patients with COVID-19. A systematic review conducted by Liu et al. showed high-dose intravenous immunoglobulin (0.4–1.0 g/kg/day) may have a decreased risk of mortality in severe COVID-19 patients than patients with usual care [27]. In contrast, Salehi et al. reported that the use of intravenous immunoglobulin did not reduce mortality in critically ill patients with COVID-19 without reducing the mortality rate [28]. This controversy in results could be attributed to differences in disease severity, dosage, and timing of intravenous immunoglobulin treatment that affected its effects. Future work is needed to identify the appropriate dosage, timing, and subgroups of patients for treating severe and critical COVID-19 patients.

Previous clinical trials had also reported conflicting outcomes regarding the efficacy of therapeutic doses of LMWH in COVID-19 patients, with some studies manifesting beneficial effects and others showing no difference between therapeutic and prophylactic doses of LMWH. A randomized trial has indicated that therapeutic-dose anticoagulants may not provide any benefit to critically ill COVID-19 patients, which is consistent with our findings. The timing of anticoagulation initiation and the effect of anticoagulation may vary depending on the severity of the illness, which could be a possible explanation for the variability in outcomes reported previously [11].

Venous and arterial thromboembolism is commonly seen in severe/critically ill COVID-19 patients and is caused by platelet activation. Antiplatelet treatment can not only halt thrombosis but also alleviate the inflammatory response in these patients [29]. However, the results of published studies on the effects of aspirin and P2Y12 antagonists (clopidogrel, ticagrelor) on hospitalized patients with COVID-19 are still controversial [30]. In our report, antiplatelet agents were found to have a 90.3% probability of improving the 60-day survival rate, although there was no significant improvement in HRQoL among survivors at two months. The possible reason for this discrepancy could be whether there is a difference between patients who received treatment with antiplatelet agents before admission and those who were assigned to antiplatelet agents after admission. We need more randomized clinical trials with larger samples to explore whether pre-existing or other antiplatelet agents might be beneficial in COVID-19 infection.

There are clinical trials confirming that dexamethasone can lower in-hospital mortality among those who were receiving either IMV or oxygen alone [13,19]. However, in our study, glucocorticoid therapy showed a 26.1% probability of a 60-day survival rate and a 30.5% probability of HRQoL improvement, indicating it did not improve survival or HRQoL. Of note, the patients who received glucocorticoid treatment were older than 70 years in our study, and this age difference may have contributed to the lack of efficacy of the regimen [23]. Further research is needed to elucidate the efficacy of glucocorticoid therapy in COVID-19 patients of different age groups.

To the best of our knowledge, this is the first real-world observational research that has reported on the effect of composite treatments for COVID-19 on longer-term mortality, HRQoL, and disability in severely and critically ill patients after the ending of the dynamic zero-COVID policy in the Chinese population. This is noteworthy as recent studies have indicated that drugs such as Paxlovid and remdesivir did not show a significant reduction in the risk of all-cause mortality in severely or critically ill adult patients with SARS-CoV-2 infection [8,21]. As the first domestic oral anti-COVID-19 drug launched in China, Azvudine has already been included in the national Guidelines on the Diagnosis and Treatment of COVID-19 (10th Trial Edition) for the treatment of adult patients with COVID-19 infections [31]. Our results support the inclusion of Azvudine in the guideline for treating severely and critically ill COVID-19 patients and highlight its potential in developing areas where access to Paxlovid is restricted.

There are some limitations to our study that should be addressed. Firstly, the duration of follow-up may not be long enough to fully evaluate the long-term effectiveness of treatment in each domain. Secondly, the results may be influenced by the effects of interventions on different variants of COVID-19 and different vaccination statuses. Thirdly, we were unable to collect baseline HRQoL and disability scores during the severe or critical stage of patients upon admission, which limits our ability to assess changes in these scores over time. Fourthly, the number of patients who received IL-6 receptor antagonist in the immune modulation domain was small, which may have introduced bias into the survival analysis. Fifthly, it is also important to acknowledge the potential limitations of the data collection method used in the study, which involved face-to-face or telephone interviews with participants, their relatives, or healthcare professionals in hospital settings. One potential source of bias is the reliance on participants or caregivers recall during the interviews, which may introduce memory biases or inaccuracies in reporting. Finally, because our study was conducted in a single center in China, the results may not be generalizable to other settings or populations. However, we continue to follow up with these discharged patients and collect information on their survival rate and functional status to determine whether the observed effects are maintained over the long term.

## Figures and Tables

**Figure 1 microorganisms-11-01859-f001:**
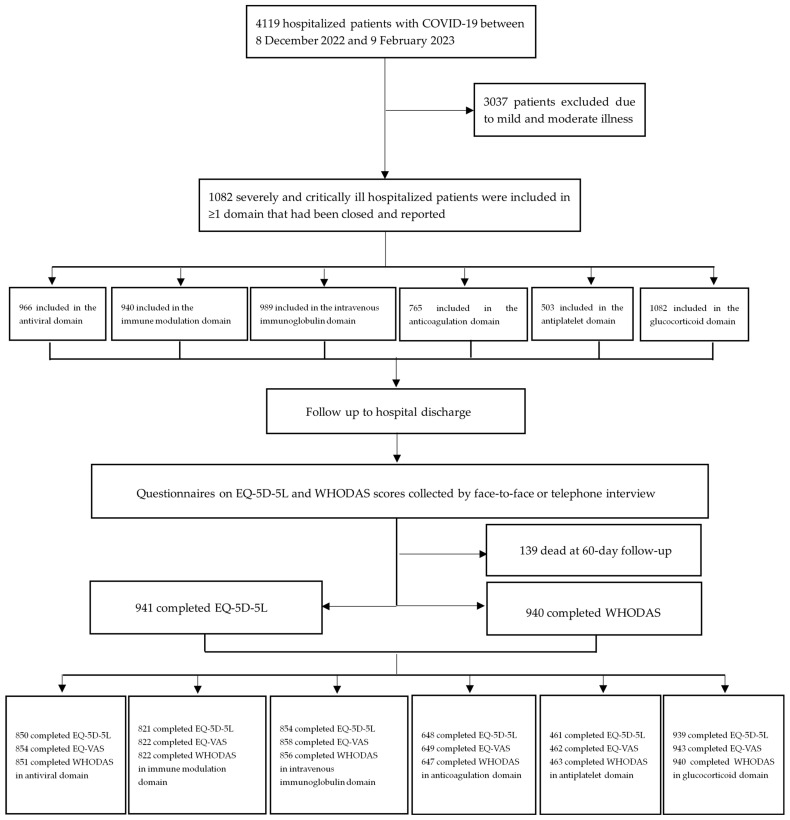
**Flow chart of the study participants**.

**Figure 2 microorganisms-11-01859-f002:**
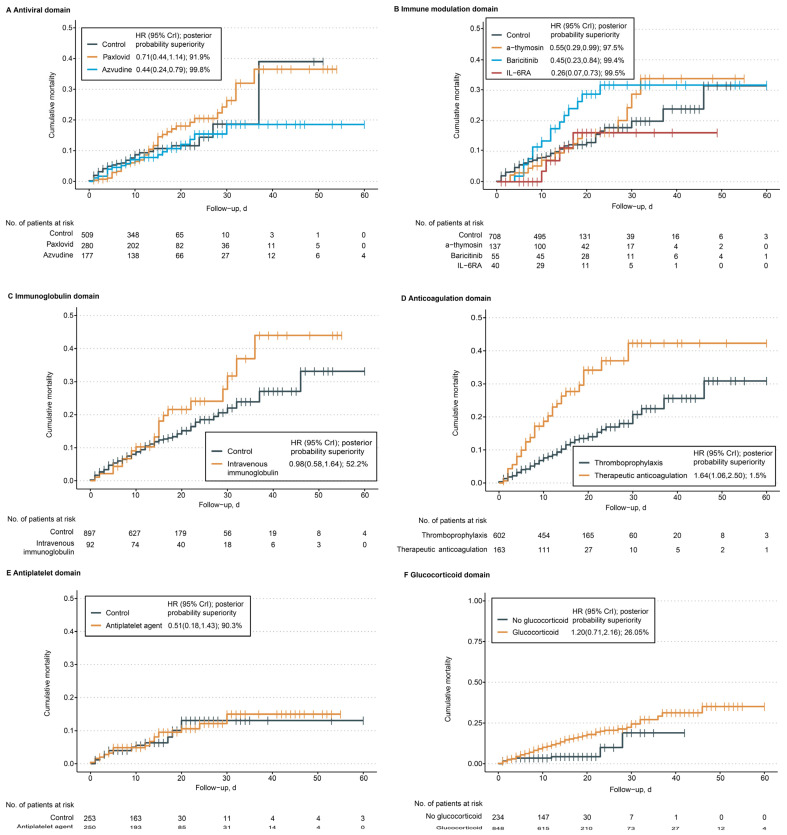
**Kaplan**–**Meier curves for mortality through 60 days.** Notes: The probability of superiority of each active intervention to control for 60-day mortality is reported from the fully adjusted Bayesian model (adjusting for patient age, sex, smoking, disease severity, preexisting conditions, respiratory supports, and other treatments from other domains). Censored participants are indicated with vertical tick marks. CrI indicates a credible interval.

**Figure 3 microorganisms-11-01859-f003:**
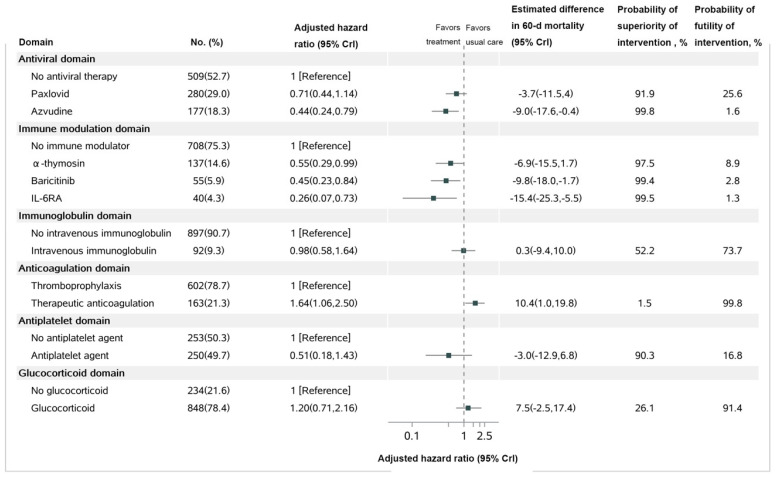
**Mortality at 60 days.** Notes: IL−6RA, interleukin−6 receptor antagonist; Hazard ratios <1 indicate improved survival, and hazard ratios > 1 indicate worsened survival. The difference in 60-day mortality is determined from the 60-day mortality rates, which are estimated from the primary analysis model. For each domain, day 60 mortality rates are estimated for the population of patients divided within that domain based on their baseline covariates and the estimated model parameters. For each patient within the domain population, separate survival curves are predicted, assuming the patient received each intervention within the domain. The mean of the survival curves was taken across patients to summarize the mean survival for each intervention within the domain population. The probability of superiority (hazard ratio < 1) and futility (hazard ratio > 0.83) is computed from a Bayesian piecewise exponential model using the posterior distribution.

**Figure 4 microorganisms-11-01859-f004:**
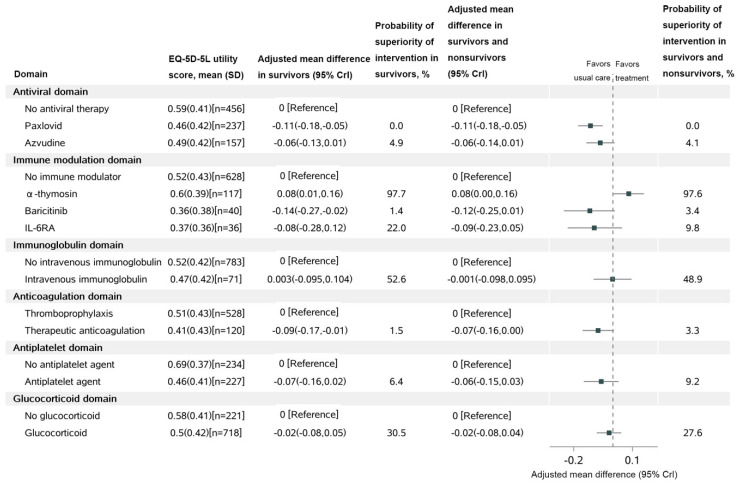
**Health**–**related quality of life at 60 days.** Notes: IL−6RA: IL−6 receptor antagonist; The probability of superiority and adjusted mean difference are computed from the posterior distribution of a bayesian 2-part/mixture model that multiplies imputes 5-level EuroQol-5 Dimension (EQ-5D-5L) utility scores using patients’ baseline covariates. For patients who were known or imputed to be alive at 60 days, a value of EQ-5D-5L is multiply imputed from the continuous component of the 2-part/mixture model. For patients who were imputed as dead by 60 days, EQ-5D-5L was set to 0, and they did not contribute to the analysis of EQ-5D-5L in survivors.

**Table 1 microorganisms-11-01859-t001:** **Baseline characteristics of patients in 1 or more domains enrolled.**

Characteristic		No./Total No. (%)
No.		1082
Age, median (IQR) [No.], y		78.0 (70.0–85.0) [*n* = 1082]
<65		168 (15.5)
65–79		441 (40.8)
80–89		348 (32.2)
≥90		125 (11.6)
Male sex, *n* (%)		661 (61.1)
Female sex, *n* (%)		421 (38.9)
BMI, median (IQR)		23.4 (21.0,26.0) [*n* = 667]
Smoking history	Smoker	140 (13.0) [*n* = 1080]
	Non-smoker	940 (87.0) [*n* = 1080]
Clinical symptoms	Fever	840 (77.6)
	Cough	891 (82.5) [*n* = 1080]
	Sore throat	435 (40.5) [*n* = 1075]
	Sputum production	837 (77.6) [*n* = 1079]
	Olfactory loss	355 (33.0) [*n* = 1077]
	Fatigue	745 (69.2) [*n* = 1076]
	Dizziness and Headache	337 (31.4) [*n* = 1074]
	Shortness of breath	758 (70.3) [*n* = 1079]
	Diarrhea	155 (14.5) [*n* = 1072]
Vaccination status	unvaccinated	705 (65.3) [*n* = 1080]
	1 dose	82 (7.6) [*n* = 1080]
	2 doses	121 (11.2) [*n* = 1080]
	3 doses	172 (15.9) [*n* = 1080]
Disease severity	Severe	966 (89.3)
	Critical	116 (10.7)
Preexisting conditions	Diabetes	437 (40.4)
	Hypertension	776 (71.7)
	Respiratory disease	140 (12.9)
	Coronary heart disease	276 (25.5)
	Cerebrovascular disease	220 (20.3)
	Chronic kidney disease	142 (13.1)
	Malignancy	154 (14.2)
	Hepatitis B virus infection	9 (0.8)
	Immunosuppressive disease	81 (7.5)
	Endocrine disease	11 (1.0)
	Liver disease	61 (5.6)
Time to hospitalization median (IQR)	From onset of symptoms to admission, d	7.0 (4.0, 10.0)
Respiratory support, No. (%)	None/supplemental oxygen supply	864 (79.9)
	High-flow nasal cannula	92 (8.5)
	Noninvasive ventilation only	87 (8.0)
	Invasive mechanical ventilation	33 (3.1)
Laboratory findings	PaO_2_/FiO_2_ (mmHg) [400–500]	297.7 (202.3, 403.6) [*n* = 815]
	WBC (×10^9^/L) [3.5–9.5]	5.2 (3.9, 7.9) [*n* = 1079]
	Hb (g/L) [130–175]	121.0 (108.0, 134.0) [*n* = 1078]
	ESR (mm/h) [<15]	42.0 (20.0, 78.0) [*n* = 482]
	CRP (mg/L) [0–8]	29.2 (6.1, 72.8) [*n* = 1058]
	Ferritin (ng/mL) [4.63–204]	543.6 (310.9, 987.8) [*n* = 327]
	LDH(IU/L) [50–240]	203.0 (127.0, 280.0) [*n* = 241]
	PLT (×10^9^/L) [125–350]	168.0 (124.0, 234.0) [*n* = 1078]
	D-dimmer (mg/L) [0–0.5]	1.0 (0.5, 2.2) [*n* = 1041]
	ALT (IU/L) [15–50]	22.0 (15.0, 36.0) [*n* = 1076]
	AST (IU/L) [17–59]	26.0 (18.0, 38.0) [*n* = 1076]
	Scr (umol/L) [46–92]	76.0 (61.3, 102.6) [*n* = 1070]
	e-GFR (ml/min) [90–120]	76.9 (55.1, 89.4) [*n* = 1043]
	Albumin(g/L) [35–53]	33.0 (30.0, 37.0) [*n* = 1040]
	cTnI(ng/mL) [≤0.1]	0.02 (0.01, 0.02) [*n* = 1003]
	Pro-BNP (pg/mL) [<589]	381.9 (125.2, 1074.4) [*n* = 1007]
	Total bilirubin (umol/L) [3.4–20.5]	10.9 (8.1, 15.0) [*n* = 1061]
	Lactate (mmol/L) [0.7–2.1]	1.8 (1.4, 2.4) [*n* = 272]
	Lymphocyte (×10^9^/L) [1.1–3.2]	0.9 (0.6, 1.3) [*n* = 1078]
	IL-6 (pg/mL) [0–10]	12.0 (4.8, 39.1) [*n* = 513]
	CD4 cell (cells/uL) [550–1440]	226.0 (37.0, 510.0) [*n* = 269]
	CD8 cell (cells/uL) [320–1250]	155.0 (25.0, 291.0) [*n* = 269]
	Potassium (mmol/L) [3.5–5.1]	4.05 (3.70, 4.50) [*n* = 1074]
	Sodium (mmol/L) [135–147]	139.0 (136.0, 141.0) [*n* = 1074]
	Chloride (mmol/L) [98–107]	106.0 (102.0, 109.0) [*n* = 1074]

Abbreviations: IQR, interquartile range; BMI, body mass index; WBC, white blood cell; Hb, hemoglobin; ESR, erythrocyte sedimentation rate; CRP, C reactive protein; LDH, lactic acid dehydrogenase; PLT, platelet; PCT, procalcitonin; ALT, alanine aminotransferase; AST, aspartate aminotransferase; Scr, serum creatinine; e-GFR, estimated glomerular filtration rate; cTnI, cardiac troponin I; BNP, B-type natriuretic peptide; IL, interleukin; CD, cluster of differentiation.

**Table 2 microorganisms-11-01859-t002:** **Domain-specific inclusion and exclusion criteria** (15, 23).

Domain	Inclusion Criteria	Exclusion Criteria
**Antiviral**	Active COVID-19 infection with cycle threshold (Ct) value < 30 times	30 mL/min ≤ creatinine clearance < 60 mL/min Severe liver dysfunction (Child-Pugh C) Known or suspected pregnancy will result in exclusion from any intervention that include Paxlovid or Azvudine Known hypersensitivity to an agent specified as an intervention in this domain will exclude a patient from receiving that agent Known HIV infection will exclude a patient from receiving Paxlovid or Azvudine Known hypersensitivity to Paxlovid or Azvudine Receiving Salmeterol, Rifampicin, Tacrolimus, Sirolimus, Domperidone, Simvastatin, Rivaroxaban, Estazolam, Atorvastatin, Amiodarone, Propafenone, or Carbamazepine as usual medications prior to this hospitalization or any administration of drugs mentioned above within 72 h prior to assessment of eligibility will exclude a patient from receiving Paxlovid
**Immune modulation**	Active COVID-19 infection The level of IL-6 in serum was 100 pg/mL and more (10 times more than upper limit of reference) C-reactive protein ≥ 75 mg/L	Patient has already received any dose of one or more of any form of α-thymosin, baricitinib, or Tocilizumab during this hospitalization or is on long-term therapy with any of these agents prior to this hospital admission Known condition or treatment resulting in ongoing immune suppression, including neutropenia, prior to this hospitalization The treating clinician believes that participation in the domain would not be in the best interests of the patient Known hypersensitivity to an agent specified as an intervention in this domain Known or suspected pregnancy will result in exclusion from α-thymosin, Tocilizumab, and baricitinib interventions. A baseline alanine aminotransferase or an aspartate aminotransferase that is more than five times the upper limit of normal will result in exclusion from receiving Tocilizumab A baseline platelet count < 50 × 10^9^/L or neutrophil < 0.5 × 10^9^/L will result in exclusion from receiving Tocilizumab Active tuberculosis, malignant tumor, thrombus, or pregnancy will result in exclusion from receiving baricitinib Patients who suffered from thymoma or who received organ transplantation will result in exclusion from receiving baricitinib
**Intravenous immunoglobulin**	Active COVID-19 infection	More than 14 days have elapsed since hospital admission Patient has already received treatment with any non-trial prescribed antibody therapy (monoclonal antibody, hyperimmune immunoglobulin, or convalescent plasma) intended to be active against COVID-19 during this hospital admission The treating clinician believes that participation in the domain would not be in the best interests of the patient Known hypersensitivity to an agent specified as an intervention in this domain Known previous history of transfusion-related acute lung injury Known objection to receiving plasma products
**Antiplatelet**	Pre-exiting disorders need antiplatelet before COVID-19 infection Recommended for all patients with severe and critical illness	Clinical or laboratory bleeding risk or both that is sufficient to contraindicate antiplatelet therapy Patient is already receiving antiplatelet therapy or non-steroidal anti-inflammatory drug (NSAID), or a clinical decision has been made to commence antiplatelet or NSAID therapy Patients otherwise eligible for the therapeutic anticoagulation domain will be excluded from the antiplatelet domain if age is more than 75 years Creatinine clearance < 30 mL/min, or receiving renal replacement therapy or ECMO Known hypersensitivity to an agent specified as an intervention in this domain will exclude a patient from receiving that agent Known or suspected pregnancy will result in exclusion from the P2Y12 inhibitor intervention Administration or intention to administer Paxlovid will result in exclusion from the P2Y12 inhibitor intervention for those who are using ticagrelor as the P2Y12 inhibitor
**Anticoagulation**	Pre-exiting disorders need anticoagulation therapy beforeCOVID-19 infection Patients otherwise eligible for the therapeutic antiplatelet domain will be included if they already have received Paxlovid	Clinical or laboratory bleeding risk, or both, that is sufficient to contraindicate therapeutic anticoagulation, including the intention to continue or commence dual anti-platelet therapy Known or suspected previous adverse reaction to low molecular weight heparin (LMWH), including heparin-induced thrombocytopenia (HIT). Patients with a platelet count of less than 50 × 10^9^/L, hemoglobin level below 80 g/L, a bleeding history within the past 30 days, or a creatinine clearance of less than 30 mL/min will be excluded from the therapeutic anticoagulation group Patients with a platelet count of less than 50 × 10^9^/L, a recent history of brain bleeding, or active bleeding who need more than 400 mL of blood transfusion, or a creatinine clearance of less than 30 mL/min will be excluded from the thromboprophylaxis group
**Glucocorticoid**	PaO_2_/FiO_2_ and imaging of Chest CT deteriorated over time during hospitalization	Known hypersensitivity to dexamethasone or methylprednisolone An indication to prescribe systemic glucocorticoids for a reason that is unrelated to the current episode, such as chronic corticosteroid use before admission, acute severe asthma, or suspected or proven *Pneumocystis jiroveci* pneumonia

**Table 3 microorganisms-11-01859-t003:** **Day 60 EQ VAS results.**

	*n*	Mean (SD)	Median (IQR)
**Antiviral domain**			
No antiviral therapy	456	69.5 (14.9)	70 (60–80)
Paxlovid	241	64.4 (16)	65 (50–75)
Azvudine	157	67.3 (15.4)	70 (60–80)
**Immune modulation domain**			
No immune modulator	629	67.8 (15.9)	70 (60–80)
α-thymosin	116	68.6 (14.8)	70 (60–80)
Baricitinib	39	65.1 (12.7)	65 (55–75)
IL-6 receptor antagonist	38	60 (14.9)	52.5 (50–70)
**Immunoglobulin domain**			
No immunoglobulin	788	67.3 (15.8)	70 (60–80)
Intravenous immunoglobulin	70	65.4 (14)	70 (50–75)
**Anticoagulation domain**			
Thromboprophylaxis	528	67.4 (15.1)	70 (55–80)
Therapeutic anticoagulation	121	62 (16.3)	60 (50–70)
**Antiplatelet domain**			
No antiplatelet agent	234	72.5 (15.1)	75 (60–85)
Antiplatelet agent	228	65.3 (13.8)	65 (55–75)
**Glucocorticoid domain**			
No glucocorticoid	224	69.1 (15.3)	70 (60–80)
Glucocorticoid	719	66.4 (15.6)	70 (55–80)

SD: standard deviation; IQR: interquartile range.

**Table 4 microorganisms-11-01859-t004:** **Day 60 disability categories.**

	Complete Disability *n*/N (%)	Severe Disability *n*/N (%)	Moderate Disability *n*/N (%)	Mild Disability *n*/N (%)	No Disability *n*/N (%)
**Antiviral domain**					
No antiviral therapy	1/458 (0.2)	175/458 (38.2)	112/458 (24.5)	155/458 (33.8)	15/458 (3.3)
Paxlovid	3/236 (1.3)	53/236 (22.5)	56/236 (23.7)	107/236 (45.3)	17/236 (7.2)
Azvudine	1/157 (0.6)	39/157 (24.8)	45/157 (28.7)	66/157 (42)	6/157 (3.8)
**Immune modulation domain**					
No immune modulator	2/629 (0.3)	201/629 (32)	144/629 (22.9)	249/629 (39.6)	33/629 (5.2)
α-thymosin	2/117 (1.7)	35/117 (29.9)	38/117 (32.5)	38/117 (32.5)	4/117 (3.4)
Baricitinib	0/40 (0)	9/40 (22.5)	5/40 (12.5)	25/40 (62.5)	1/40 (2.5)
IL-6 receptor antagonist	0/36 (0)	5/36 (13.9)	6/36 (16.7)	23/36 (63.9)	2/36 (5.6)
**Immunoglobulin domain**					
No immunoglobulin	3/785 (0.4)	236/785 (30.1)	190/785 (24.2)	323/785 (41.1)	33/785 (4.2)
Intravenous immunoglobulin	2/71 (2.8)	20/71 (28.2)	16/71 (22.5)	29/71 (40.8)	4/71 (5.6)
**Anticoagulation domain**					
Thromboprophylaxis	4/527 (0.8)	162/527 (30.7)	125/527 (23.7)	209/527 (39.7)	27/527 (5.1)
Therapeutic anticoagulation	0/120 (0)	22/120 (18.3)	30/120 (25)	59/120 (49.2)	9/120 (7.5)
**Antiplatelet domain**					
No antiplatelet agent	2/235 (0.9)	128/235 (54.5)	53/235 (22.6)	46/235 (19.6)	6/235 (2.6)
Antiplatelet agent	0/228 (0)	53/228 (23.2)	61/228 (26.8)	103/228 (45.2)	11/228 (4.8)
**Glucocorticoid domain**					
No glucocorticoid	0/223 (0)	74/223 (33.2)	57/223 (25.6)	81/223 (36.3)	11/223 (4.9)
Glucocorticoid	5/717 (0.7)	205/717 (28.6)	175/717 (24.4)	298/717 (41.6)	34/717 (4.7)

**Table 5 microorganisms-11-01859-t005:** **Day 60 disability comparison.**

	N (Survivors at Day 60)	WHODAS Score, Median (IQR) [N]	Adjusted Odd Ratio (95% CrI)	*p* Value
**Antiviral domain**				
No antiviral therapy	458	86.5 (59–132)	1.00 (reference)	
Paxlovid	236	109 (73–147)	0.45 (0.32, 0.62)	<0.001
Azvudine	157	102 (71–141)	0.68 (0.47, 0.97)	0.034
**Immune modulation domain**				
No immune modulator	629	98 (65–144)	1.00 (reference)	
α-thymosin	117	92 (67–124)	1.43 (0.94, 2.18)	0.095
Baricitinib	40	135.5 (79–156)	0.68 (0.33, 1.37)	0.282
IL-6 receptor antagonist	36	120 (99–133)	0.75 (0.25, 2.30)	0.621
**Immunoglobulin domain**				
No immunoglobulin	785	99 (65–141)	1.00 (reference)	
Intravenous immunoglobulin	71	102 (67–143)	1.68(1.00, 2.82)	0.052
**Anticoagulation domain**				
Thromboprophylaxis	527	99 (65–141)	1.00 (reference)	
Therapeutic anticoagulation	120	114.5 (81–156)	0.65 (0.44, 0.96)	0.03
**Antiplatelet domain**				
No antiplatelet agent	235	69 (56–102)	1.00 (reference)	
Antiplatelet agent	228	107.5 (75–144.5)	0.62 (0.38, 1.02)	0.06
**Glucocorticoid domain**				
No glucocorticoid	223	96 (61–129)	1.00 (reference)	
Glucocorticoid	717	101 (68–145)	1.16 (0.84, 1.60)	0.355

Notes: IQR: interquartile range; CrI: credible interval Odds ratio parameters are for the analysis of WHODAS disability category; An OR < 1 indicates reduced disability. *p* value is computed from an ordinal mixture mode.

## Data Availability

Raw data used in this study, including de-identified patient metadata and test results, are available upon request. Reasonable requests to access the datasets should be directed to the corresponding authors.

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
