# Peer review of "Composite Interventions on Outcomes of Severely and Critically Ill Patients with COVID-19 in Shanghai, China"

_microorganisms, 2023, doi:10.3390/microorganisms11071859_

Round 1
Reviewer 1 Report
This is a very interesting paper with a lot of data
I have a few minor comments for the authors to adress
In Abstract, I am not sure Paxlovid had significnt result on survival, please check. Also check results for glucocorticoid. Dont put comma before and
Line 53, green mark - remove
Line 59 missing space (check through the manuscript)
Line 61 - specify year
Line 106 - space extra
I do not understand why would patients in antiviral not receive antivirals, please clarify in methods how you divided patients. Also I would suggest you keep to one terminology in the manuscript i.e. mechanism of action of drugs or drug name but not mix and match both
Consider increasing font and resolution of figure 1.
Table 1 is completely unclear why you have number of patients per domain (single) and then all, consider presenting this differently.
Table 2 is missing inclusion criteria related to medications...
Data was collected by face-to-face or telephone interview with the participants, their relatives, or health care professional in our hospital - I think that this method of collection may yield substantial bias and variations so please comment on this in discussion. Also repeated statistical analysis creates space for a false positive, and cesored data i.e. for deceased patients and how their HRQoL was treated should be clarified in the methods.
Resolution in Figure 2 also needs to be improved
Reviewer 2 Report
The study is of interest as it is the first real-world observational trial to study a combination treatment of COVID patients that includes Paxlovid and remdesivir.
The number of patients is large enough to be statistically significant.
As a criticism, the study was conducted in a single country, which may offer important racial differences, but nevertheless I think this is something the authors can think about for a second version.
On the other hand, the tables need to be improved as they are difficult to understand and visualise.
